# Improvement of the Surface Properties of Polyether Ether Ketone via Arc Evaporation for Biomedical Applications

**DOI:** 10.3390/ma16082990

**Published:** 2023-04-09

**Authors:** Alexander Y. Fedotkin, Igor O. Akimchenko, Tuan-Hoang Tran, Artur R. Shugurov, Evgeniy V. Shesterikov, Anna I. Kozelskaya, Sven Rutkowski, Sergei I. Tverdokhlebov

**Affiliations:** 1Weinberg Research Center, School of Nuclear Science & Engineering, Tomsk Polytechnic University, 30, Lenin Avenue, 634050 Tomsk, Russia; 2Institute of Strength Physics and Materials Science, Siberian Branch of the Russian Academy of Sciences, 2/4, pr. Akademicheskii, 634055 Tomsk, Russia; 3Nanotechnology Research and Education Center, Tomsk State University of Control Systems and Radioelectronics, 40, Lenin Avenue, 634050 Tomsk, Russia

**Keywords:** 3D-printing, polyether ether ketone, arc evaporation, titanium coatings, roughness, wettability

## Abstract

Polyether ether ketone is a bioinert polymer, that is of high interest in research and medicine as an alternative material for the replacement of bone implants made of metal. The biggest deficit of this polymer is its hydrophobic surface, which is rather unfavorable for cell adhesion and thus leads to slow osseointegration. In order to address this drawback, 3D-printed and polymer extruded polyether ether ketone disc samples that were surface-modified with titanium thin films of four different thicknesses via arc evaporation were investigated and compared with non-modified disc samples. Depending on the modification time, the thickness of the coatings ranged from 40 nm to 450 nm. The 3D-printing process does not affect the surface or bulk properties of polyether ether ketone. It turned out that the chemical composition of the coatings obtained did not depend on the type of substrate. Titanium coatings contain titanium oxide and have an amorphous structure. Microdroplets formed on the sample surfaces during treatment with an arc evaporator contain a rutile phase in their composition. Surface modification of the samples via arc evaporation resulted in an increase in the arithmetic mean roughness from 20 nm to 40 nm for the extruded samples and from 40 nm to 100 nm for the 3D-printed samples, with the mean height difference increasing from 100 nm to 250 nm and from 140 nm to 450 nm. Despite the fact that the hardness and reduced elastic modulus of the unmodified 3D-printed samples (0.33 GPa and 5.80 GPa) are higher than those of the unmodified extruded samples (0.22 GPa and 3.40 GPa), the surface properties of the samples after modification are approximately the same. The water contact angles of the polyether ether ketone sample surfaces decrease from 70° to 10° for the extruded samples and from 80° to 6° for the 3D-printed samples as the thickness of the titanium coating increases, making this type of coating promising for biomedical applications.

## 1. Introduction

Titanium and titanium alloys are already traditional materials for the manufacture of implants. These materials are popular for intrabody implants due to their relatively low density, excellent biocompatibility, good mechanical properties, high corrosion resistance, low allergy-causing properties, and good chemical durability [1,2,3]. However, a major disadvantage of titanium implants is their high modulus of elasticity, which leads to a shielding effect of tensions that results in bone resorption due to a mismatch between the elastic modulus of cortical bones (10–17 GPa) [4] and that of titanium (105–114 GPa) [5]. The use of polyether ether ketone (PEEK) can solve this problem since the elastic modulus of this polymer is close to that of natural bone [6]. In addition, compared to titanium, PEEK has advantages in terms of manufacturing and processing, which in this case is mainly done by 3D printing [7,8]. However, the main disadvantage of PEEK is its bioinertness, which results in low bioactivity and prevents its integration into bone tissue [6,9]. Surface modification of PEEK implants can solve this problem and improve the adhesion and proliferation of osteoblasts, as well as the differentiation of stem cells into osteoblasts.

There are several strategies for surface modification of PEEK: chemical treatment, physical treatment, and the deposition of bioactive coatings [10]. Recently, much attention has been paid to the formation and application of bioactive coatings on PEEK implants. At the same time, a wide range of coating methods is available: electrochemical deposition [11], magnetron sputtering [12], cathodic arc evaporation (CAE) [13] and chemical deposition [14]. However, electrochemical deposition methods for surface coating of PEEK cannot be applied due to the non-metallic properties of the polymer. PEEK can be modified by magnetron sputtering, but the resulting coatings are smooth. At the same time, CAE can be used to form coatings with higher roughness [15], which is beneficial for cell adhesion. The disadvantage of this method is the presence of microdroplets in the flow of evaporated particles, which is disadvantageous for the formation of smooth surface coatings and the adhesion properties [15]. However, this can be advantageous for the surface modification of implants, as it offers the possibility to control the roughness of the material surface by adjusting the processing time [16]. The most important surface modifications of PEEK are coatings based on hydroxyapatite [17,18,19,20,21] and titanium [19,22,23,24], which are known for their osteoconductive properties.

In this study, to investigate the surface properties of PEEK, titanium-based coatings were deposited on PEEK samples by cathodic arc evaporation. Moreover, this work also investigated the influence of 3D printing on the surface properties of coated PEEK samples. Therefore, two types of disk-shaped PEEK substrates were used: a sliced one from a solid polymer rod and a 3D-printed one. The results of the study may be useful for the development of 3D titanium-coated PEEK scaffolds, which offer higher roughness and better surface wettability. In addition, the obtained coatings can be used as an adhesive layer for the subsequent application of a calcium phosphate coating to increase the biocompatibility of PEEK implants.

## 2. Materials and Methods

### 2.1. Substrate Preparation

In this study, two types of polyether ether ketone (PEEK) samples were used as substrates for surface modification and investigation: polished PEEK samples cut from extruded polymer bars (polished PEEK) and 3D-printed PEEK (3D–PEEK) samples. These two sample types were chosen in order to evaluate not only the effect of the arc evaporation process on the surface and bulk properties of the substrates but also the potential differences that were related to the fused deposition modeling (FDM) 3D printing process. The PEEK substrates were cut from extruded polymer bars, ground, and polished (Figure 1). Grinding and polishing processes were carried out using a precision grinding and polishing machine (Unipol-802, Zhengzhou TCH Instrument Co. Ltd., Zhengzhou, China). As it can be seen in Figure 1, 3D–PEEK samples were manufactured using a 3D-printer (Creatbot Peek 300, Creatbot, Zhengzhou, China). The side that rested on the 3D printer’s printing table was used as the front side of the samples without grinding and polishing. All samples were cleaned in isopropanol using an ultrasonic bath. All substrate samples were prepared in the form of discs with a diameter of 10 mm and a thickness of 2 mm (Figure 1).

### 2.2. Surface Modification Parameters

Coatings were deposited using a customized vacuum deposition unit (Figure 1) developed in Tomsk Polytechnic University based on the vacuum chamber of an UVN-74 of the company AO-KVARTZ (Kaliningrad, Russia). The following parameters for the arc evaporation process were applied: working gas: argon (Ar), working pressure: ~5.5 Pa, current: 35 A, frequency: 10 Hz, distance between target and substrate: 140 mm. The disk samples of extruded PEEK and 3D-printed PEEK were surface-modified for 1 (PEEK–1 and 3D–PEEK–1), 3 (PEEK–3 and 3D–PEEK–3), 5 (PEEK–5 and 3D–PEEK–5), and 10 min (PEEK–10 and 3D–PEEK–10). A pause was taken after each minute of the arc evaporation process to avoid melting the polymer samples.

### 2.3. Investigation Methods

A schematic overview of the investigation methods used is shown in Figure 1. The morphology of the sample surfaces was investigated via atomic force microscopy (AFM) using an NTEGRA PRIMA (NT-MDT, Moscow region, Russia) in the semicontact mode. For that, a NSG01 probe (NT-MDT, Moscow region, Russia) with a force constant in the range of 5.1 N/m and a resonant frequency in the range of 87 to 230 kHz has been used. AFM micrographs were processed using Gwyddion software (version 2.62, gwyddion.net, Brno, Czech Republic), and roughness values were determined using the same software. This article shows AFM micrographs with the sizes of 40 × 40 μm^2^ and 3 × 3 μm^2^.

The coating thickness, deposition rate, and elemental composition of the samples and the distribution of elements on their surfaces were evaluated using a scanning electron microscope (Quanta 200 3D, FEI Company, Hillsboro, OR, USA) equipped with an energy-dispersive X-ray spectroscope (EDX; JSM-5900LV, JEOL Ltd., Akishima, Japan). SEM-EDX investigations were performed under low vacuum with an accelerating voltage of 20 kV. SEM micrographs were obtained in a high vacuum at a magnification of ×10,000.

Chemical characterization of the fabricated coatings has been investigated by X-ray photoelectron spectroscopy (XPS; NEXSA, Thermo Fisher Scientific, Waltham, MA, USA). This device was equipped with a monochromatic Al-K alpha X-ray source operating at 1486.6 eV. The measured survey spectra were taken at a pass energy of 200.0 eV. In turn, the high-resolution spectra were obtained at 50.0 eV with a step range of 0.1 eV. The examined sample surfaces had an area of 400.0 μm^2^. The studies were carried out in ultrahigh vacuum at a pressure of 1.0 × 10^−5^ Pa and at room temperature. When using the electron-ion compensation system, the argon partial pressure equaled 1.0 × 10^−3^ Pa.

X-ray diffraction (XRD) measurements were performed on an X-ray diffractometer (XRD-6000, Shimadzu, Kyoto, Japan) with Cu Kα radiation (λ = 0.154 nm). The X-ray generator was operated at 43 kV and 150 mA. Each diffraction pattern was recorded over a 2*θ* angle range from 10° to 55° with a step size of 0.05°. The ICDD PDF4+ database was used for phase identification.

Raman spectroscopies were performed using a confocal Raman microscope (NTEGRA Spectra, NT-MDT, Zelenograd, Moscow region, Russia). The Raman microscope was equipped with a diode laser with a wavelength of 532 nm. The laser beam was focused on the sample using a ×100 objective (M Plan Apo HR 100X, Mitutoyo, Aurora, IL, USA) with a numerical aperture of 0.7. The signal was collected using an electron-multiplying charge-coupled detector (EMCCD) (Newton 971, Andor Technology Ltd., Belfast, Northern Ireland) cooled down to −65 °C. The PEEK samples exhibited fluorescent properties, which allowed them to be studied by Raman spectroscopy.

Wettability of the samples was determined by the sessile drop method using an EasyDrop DSA20 (Krüss, Hamburg, Germany). For this purpose, contact angles for water (WCA), glycerol (WCA), and diiodomethane (DCA) were determined. The volume of each drop was 3 mL. Additionally, the surface free energy (σ), its polar (σ_D_) and its dispersion (σ_P_) components were determined by the Owens, Wendt, Rabel, and Kaelble (OWRK) method. This investigation was conducted in accordance with the requirements of ISO 19403-2:2020.

The mechanical properties of the samples were determined by nanoindentation using a NanoTest system (Micro Materials Ltd., Wrexham, UK) operated in the load-controlled mode using a Berkovich diamond tip. For this purpose, the maximum load on the diamond tip was set at 20 mN, and the loading and unloading times were set at 20 s with a 10 s dwell time at maximum load and a 60 s dwell time at 90% unload for thermal drift correction. The hardness (*H*) and reduced elastic modulus (*E**) of the samples were determined from load vs. displacement curves using the Oliver–Pharr method [25].

The statistical significance of the results was determined using one-way analysis of variance and the Mann-Whitney U-test (Statistica 7.0, StatSoft, Tulsa, OK, USA).

## 3. Results and Discussion

Cross-sectional SEM micrographs of polished and surface-modified PEEK samples cut from an extruded polymer bar allow assessment of the thickness of the coatings formed (Figure 2a–d). It can be seen that the coating thickness increases with increasing deposition time, from 40 nm to 450 nm. The dependence of the thickness of the titanium coatings deposited by arc evaporation on time is almost linear (Figure 2e). From the graph in Figure 2e, the average deposition rate of the coatings can be estimated, which is 40 nm/min. With this result, the coating thickness can be regulated with high accuracy by varying the processing time. It should be noted that the thickness of the coatings formed under the same working parameters of the cathodic arc evaporation on the PEEK and 3D–PEEK samples is in the same range, as evidenced by the analysis of cross-sectional SEM micrographs of the coatings (see Appendix A). Photographs and SEM micrographs of the surfaces of all sample types investigated are shown in Appendix A, respectevely. These photographs show that the shape and size of the samples did not change due to the coatings. Only the color of the sample surfaces slowly changed from initially light beige (unmodified samples) to light gray, which can be seen most clearly on the samples after 10 min of deposition time (Appendix A). The polished PEEK samples (Appendix A) show small polishing traces on the surface, which disappear, when the coatings are deposited. In contrast, the surface of 3D–PEEK (Appendix A) is smooth. It is worth noting that no serious differences between the polished PEEK and 3D–PEEK samples can be seen in the SEM micrographs in Appendix A. In details, the surface of the samples are smooth, and have microdroplets that vary in size from 0.3 μm to 3 μm. The number of microdroplets on the sample surfaces increases with increased deposition time. The presence of microdroplets on the surface of the coatings formed by cathodic arc evaporation is a common phenomenon that was also observed by other authors [26,27,28].

After 1 min of surface modification, the surfaces of the PEEK–1 and 3D–PEEK–1 samples (Appendix A) differ only slightly from the unmodified samples. Individual, non-contiguous, structural titanium-based coatings of various shapes are observed on the sample surfaces. The surface of samples from the PEEK–3 and 3D–PEEK–3 groups (Appendix A) is characterized by the presence of grain-like structures over the entire surface. After 3 min of deposition time, uniform coatings are formed. With a further increase in the deposition time up to 5 min (PEEK–5 and 3D–PEEK–5, Appendix A) and 10 min (PEEK–10 and 3D–PEEK–10, Appendix A), the size of the grain-like structures on the coated surfaces and the number of inclusions increase.

EDX mapping micrographs of the surface-modified samples show that the coatings consist of titanium and oxygen (Appendix A). The presence of oxygen in the coatings can be explained in two ways: 1. during the surface modification of the PEEK substrates by cathodic arc evaporation, reactive oxygen originating from the functional groups of the PEEK molecules was formed; 2. oxidation in air after removal of the samples from the vacuum chamber. In turn, the presence of carbon in the EDX mapping micrographs may be predominantly due to the polymeric nature of the substrates. This is evidenced by the fact that with increasing deposition time, the coating thickness and the amount of carbon gradually decrease for polished PEEK samples (Appendix A) and also for 3D–PEEK samples (Appendix A). At the same time, the content of oxygen and titanium in the coatings increases. This trend proves that the coatings contain titanium oxide. Titanium is a d-block element and belongs to the transition metals. It can exist in a divalent, trivalent, or tetravalent oxidation state. Most often, titanium forms tetravalent compounds because the tetravalent oxidation state is the most stable one. Therefore, it can be assumed that titanium oxide is represented by TiO_2_. The source of titanium for the oxide formation is the flow of particles deposited by the arc evaporation process. When the titanium ions released from the targed interacting with air, a layer of titanium dioxide with a thickness of 2–5 nm is formed [29]. In this study, the coating thicknesses obtained range from 40 nm to 450 nm, depending on the deposition time. However, with an increasing deposition time, an increase in the oxygen content of the coatings is observed. Consequently, the titanium oxide content in the coatings increases with increasing deposition time, and the main source of oxygen in titanium dioxide is reactive oxygen ions that orignate from the PEEK backbone structure on the substrate surfaces caused by the working temperature in the chamber. This fact is confirmed by a continuous increase in the chamber pressure when the arc evaporator is working and the need to lower the pressure to maintain the value constant. Otherwise, the oxygen content would decrease in the deposited coatings with increasing coating thickness.

It should also be noted that the spherical structure elements (here referred to as microdroplets) described in the morphology investigation part of this study consist of titanium. At the same time, the oxygen content is significantly lower of the microdroplets than the flat part of the coating. This confirms the assumption that these objects are titanium microdroplets formed on the substrate surfaces during the cathodic arc evaporation process.

The elemental composition of the samples determined by energy-dispersive X-ray spectroscopy (EDX) in atomic percent (at.%) is shown in Figure 3 and Appendix A. As mentioned above, these are elements characteristic of the composition of the modified substrates and the evaporated target. The presence of foreign elements is not observed. However, the content of oxygen (O) and titanium (Ti) in the 3D–PEEK samples after 3 and 5 min of deposition time is lower than that in the polished PEEK. For the unmodified polished PEEK and 3D–PEEK samples, the ratio of oxygen bound to carbon (O/C) is 0.17 (Appendix A, column 5), which is close to the literature value [30]. To calculate the titanium to oxygen (Ti/O) ratio (Appendix A, column 7), the O/C ratio of 0.16/1 for pristine PEEK from the literature [30] was first used to calculate the O/C ratios of the surface-modified samples (Appendix A, column 5). The remaining oxygen, which is not bound to carbon, is bound to titanium, resulting in the Ti/O ratio given in Appendix A, column 6. As it can be seen, the obtained Ti/O ratios have similar values for the polished PEEK and 3D–PEEK samples. 

For surface-modified polished PEEK samples and surface-modified 3D–PEEK samples with a Ti coating, the survey spectra also show a Ti2p peak (Appendix A). The content of titanium is approximately the same for all samples with coatings and ranges from 24–27 at.% (Appendix A).

Figure 4 shows the high-resolution XPS spectra of all investigated samples. In the case of unmodified polished PEEK and 3D–PEEK samples, O1S and C1S peaks are observed. The content of carbon (C1s) is 84 ± 2 at.% and for oxygen (O1s) 16 ± 1 at.% for the unmodified PEEK samples and for the unmodified 3D–PEEK samples it is 86 ± 2 at.% and 14 ± 1 at.%, respectively (Appendix A). In both cases, the O/C ratio is 0.19, which is close to the value of 0.16 indicated in the literature [30].

Moreover, the Ti2p spectra (Figure 4, upper row) of the surface-modified samples show peaks characteristic of titanium with the oxidation state of +4, Ti2p_3/2_ (459.4 eV), and Ti2p_1/2_ (465.0 eV), with a distance between them of about 5.6 eV [31]. These peaks are not present in the XPS spectra of the unmodified samples (Figure 4, top row). The O1s spectra for oxygen of all surface-modified samples (Figure 4, lower row) show a peak at 530.7 eV, which corresponds to Ti–O bonds [32]. For unmodified PEEK and 3D–PEEK samples, there are peaks characteristic of PEEK at 533.1 eV and 531.0 eV, corresponding to O–C and O=C bonds, respectively [33]. From the obtained XPS results, it can be concluded that the coatings of the surface-modified samples consist of titanium dioxide (TiO_2_).

The results of the study of the morphology of the samples by SEM (Appendix A) are in agreement with those obtained by AFM (Appendix A). Examining a sample surface area of 3 × 3 μm^2^, the presence of grain-like structures is observed, which is in contrast to unmodified sample surfaces. The polished PEEK–3, PEEK–5 and PEEK–10 samples are characterized by an increase in the number and size of microdroplets, which can be seen in AFM micrographs with a scanning area of 40 × 40 μm^2^ as the deposition time increases. From the AFM micrographs of 3 × 3 μm^2^, it can be seen that the size of the grain-like structures on the coating surface also increases. The surface of the unmodified 3D–PEEK sample is characterized by a smooth surface with voids formed during the 3D-printing process by incomplete filling of voids with filament material. On the surface of 3D–PEEK samples that have been surface-modified for 1 min, no recognizable structural elements are visible on the 40 × 40 μm^2^ AFM micrographs. The beginning of the formation of grain-like structures can be seen on AFM micrographs with an area of 3 × 3 μm^2^. On the surfaces of the 3D–PEEK–3, 3D–PEEK–5 and 3D–PEEK–10 samples, an increase in the number and size of microdroplets is observed, which are detectable in the 40 × 40 μm^2^ AFM micrographs. An increase in the size of the grain-like structures is also observed on the surfaces of these samples in AFM micrographs with an area of 3 × 3 μm^2^. It can be concluded that the coating deposition process on the surfaces of polished PEEK and 3D–PEEK substrates during arc evaporation proceeds in a similar manner. Starting after 1 min of deposition time, the process of grain-like structure formation begins for both sample types. If the deposition time is further extended, the number and size of microdroplets and grain-like structures on the surface of the samples increase. All differences between the polished PEEK and 3D–PEEK samples are due to differences in the morphology of the unmodified samples.

The results of the SEM investigations confirm the results of the AFM investigations: with increasing deposition time during the arc evaporation process, the number and size of the grain-like structures and the size of the observed titanium microdroplets increase, independently of the fabrication method of the PEEK samples.

Figure 5 and Appendix A present the arithmetic mean roughness (*R_a_*) and the average height difference (*R_z_*) of the examined samples obtained by AFM. The roughness of the coatings formed on the surface of the polished PEEK samples generally increases with increasing deposition time. This result is to be expected since a feature of arc evaporation is the expansion of the droplet fraction of the evaporated target. Accordingly, an increase in the deposition time leads to an increase in the number of microdroplets deposited on the sample surfaces, which results in an increase in roughness. 

Also noteworthy is a more significant increase in *R_a_* and *R_z_* for the 3D–PEEK–5 and 3D–PEEK–10 samples (Figure 3b) compared to the polished PEEK samples (Figure 3a). This phenomenon is caused by the effect of geometric shadowing [34,35,36]. In particular, the irregularities of the 3D–PEEK samples (caused by the 3D printing process) hide parts of the sample surface from the oncoming particle stream during the coating process. In turn, the peaks of the surface irregularities are exposed to particles from all directions during the coating process, making it easier for the coatings to form there, which in turn contributes to the higher measured roughness values. This effect becomes most pronounced when the formed coating is sufficiently thick, especially after a deposition time of 5 to 10 min. Figure 4 shows the roughness (*R_a_* and *R_z_*) for a sample surface area of 40 × 40 μm^2^ of polished PEEK and 3D–PEEK samples with increasing deposition time by arc evaporation.

Figure 6 and Appendix A show the values of the wettability of the PEEK samples as water contact angles (WCA) and diiodomethane contact angles (DCA), as well as the surface energies (σ) and their polar (σ_P_) and dispersion (σ_D_) components. There is a general trend towards an increase in the wettability of all samples with increasing deposition time. According to Appendix A, the sizes of structural elements and defects on the surface of the coatings do not exceed several micrometers, which makes it possible to discuss the formation of a micro- and nanoscale relief. Consequently, the Wenzel-Deryagin and Cassie-Baxter models can be applied to describe the wettability of the samples, since the droplet size of a liquid for the contact angle measurement significantly exceeds the characteristic scale of the structural elements and defects on the coating surfaces. The Cassie-Baxter model explains the behavior of a droplet in the case of heterogeneous wetting, where surface irregularities are filled with air. According to this model, when a surface has pronounced hydrophobic properties, an energy barrier ensures that the air remains below the liquid, and the air layer displaces the liquid on the surface [37]. According to the Wenzel-Deryagin model, when wetting a hydrophilic surface, an increase in its roughness leads to a decrease in the contact angle, while for a hydrophobic surface, an increase in the roughness leads to an increase in the contact angle [38]. This model assumes homogeneous wetting, in which a liquid is wetting the entire surface of a substrate, completely filling its pits. Since a general trend of decreasing the wetting angle with increasing coating time was observed, it can be assumed that wetting of the studied coatings occurs according to the Wenzel-Deryagin model.

All surface-modified samples are characterized by large values of the surface energy (σ) in comparison with the unmodified samples (Figure 6b). At the same time, unmodified polished PEEK and unmodified 3D–PEEK samples are characterized by a pronounced dispersion component σ_D_. When the deposition time increases to 10 min, the polar component σ_P_ of the PEEK–10 samples becomes comparable to the dispersion component, σ_D_. The 3D–PEEK samples are characterized by a distinct polar component σ_P_. Moreover, the surface modification of the 3D–PEEK samples led to a decrease in the difference between the dispersion σ_D_ and polar components σ_P_ of the surface energy σ. The dependence of cell adhesion and proliferation on the polar component of the surface energy σ_P_ has been demonstrated in other studies [39,40,41]. In reference [41], the surface of an implant with a high polar component of the surface energy σ_P_ displays the highest cell density and the lowest level of inflammatory cytokines by fibroblasts. In contrast, implants with a very low polar component of the surface energy σ_P_ showed significantly higher expression of inflammatory mediators and lower cell proliferation.

According to the XRD results (Figure 7a,b), the spectra of all samples are characterized by four diffraction peaks originating from PEEK itself at *θ* = 18.8°, 20.7°, 22.7°, and 28.7°, corresponding to crystal planes with indices (110), (111), (200) and (211) [42]. However, no further peaks are present. This fact indicates either an amorphous coating structure [43] or a too low thickness of the deposited thin films for the determination of the crystal structure by XRD. To investigate the phase composition of the coatings further, silicon (Si) substrates with the same coatings as those deposited on the PEEK samples were additionally investigated using RAMAN spectroscopy.

Raman spectroscopy shows only peaks at 301, 520, and 976 cm^−1^ at the areas where no microdroplets are present and originate from the SI substrates (Figure 7c) [44]. No peaks corresponding to titanium oxide phases are present in the coatings. Therefore, it can be assumed that the coatings are amorphous. At the same time, the presence of peaks corresponding to rutile (143, 241, 433, and 610 cm^−1^) [45] present in the microdroplets (Figure 7d) for Si samples with coatings deposited for 3, 5, and 10 min, in which the amount of oxygen was lower than in the thin films, as evidenced by EDX (Appendix A). These peaks are characterized by low intensity and large peak width, indicating low crystallinity of the microdroplets.

The elemental composition of the samples is shown in Appendix A, where the elements are characteristic of the molecular composition of PEEK and the evaporated target. A similar dependence is observed for samples surface-modified by the deposition of titanium coatings both on the surface of polished PEEK and 3D–PEEK: the content of carbon decreases, the content of titanium and oxygen increase with the increasing deposition time, as it is mentioned above. The simultaneous increase in the content of titanium and oxygen with increasing deposition time indicates the presence of titanium oxide in the coatings.

In summary, the EDX, XRD, and Raman spectroscopy results show that the coatings deposited on the PEEK substrates represent an amorphous phase of titanium oxide. Microdroplets, on the other hand, are characterized by a lower oxygen content, and their structure is characterized by the presence of rutile.

Appendix A displays the mechanical properties of the unmodified PEEK samples. These data show that the values for the hardness (*H*) and the reduced Young’s modulus (*E**) are lower for the 3D–PEEK samples than for the polished PEEK samples. The lower mechanical properties of the 3D–PEEK samples are due to weak interfacial bonding of the printed structures caused by large temperature differences between the printing head and the printing table in the fused deposition modeling (FDM) printing process [46]. Therefore, one way to improve the mechanical properties is to anneal the samples in a solvent atmosphere at room temperature [46]. To reduce the degradation of mechanical properties during the FDM printing process, Yang et al. [47] proposed a novel process to control the crystallinity and mechanical properties of 3D-printed carbon fiber-reinforced PEEK (CF/PEEK) composite material via a recrystallization process. Thus, today there are many options for improving the mechanical properties of 3D–PEEK samples, which will have a positive impact on the application of 3D–PEEK materials in the field of biomedical implants.

## 4. Conclusions

The study presented here has successfully shown that the morphology of the coatings does not depend on the type of substrate used (polished polyether ether ketone disk-like samples cut from a bar or 3D-printed polyether ether ketone disk-like samples). However, the mechanical properties of 3D-printed polyether ether ketone samples are inferior to polished polyether ether ketone samples. It has been found that with increasing deposition time, the thickness of the coatings and the surface roughness are increasing. The observed phenomenon of a more pronounced increase in the arithmetic mean roughness and the average height difference for 3D-printed polyether ether ketone samples compared to polished polyether ether ketone samples is due to the effect of geometric shadowing caused by the presence of irregularities on the surface of the 3D-printed samples. Moreover, microdroplets formed on the surfaces of the coatings affect the increase in surface roughness of both types of samples. The number of microdroplets that are formed increases with increased deposition time. These microdroplets were detected using atomic force microscopy and scanning electron microscopy, and their titanium-based structure was confirmed using energy dispersive X-ray spectroscopy. The obtained coatings contain titanium oxide with an amorphous structure, while the microdroplets formed on the coating surfaces consist of rutile, which is known for a good biocompatibility. 

From the obtained results of scanning electron microscopy, X-ray photoelectron spectroscopy, X-ray diffraction, and Raman spectroscopy, the coatings on the surfaces of polished polyether ether ketone samples and 3D-printed polyether ether ketone samples have similar characteristics. For surface roughness and wettability, the measured values are slightly different. At the same time, as the duration of cathodic arc evaporation increases, the roughness and wettability of the coatings increase. Based on these results, it can be concluded that polished polyether ether ketone samples cut from extruded polymer bars can also be used as a model to study the properties of 3D-printed polyether ether ketone samples whose surface has been modified by cathodic arc evaporation. 

A significant improvement in wettability, even at low film thicknesses, and the ability to adjust roughness by adjusting the deposition time during arc evaporation make this type of polyether ether ketone coating promising for biomedical applications. The coatings prepared and investigated in this study can be used either as stand-alone coatings or as a sublayer for subsequent deposited coatings, e.g., calcium phosphate coatings.

## Figures and Tables

**Figure 1 materials-16-02990-f001:**
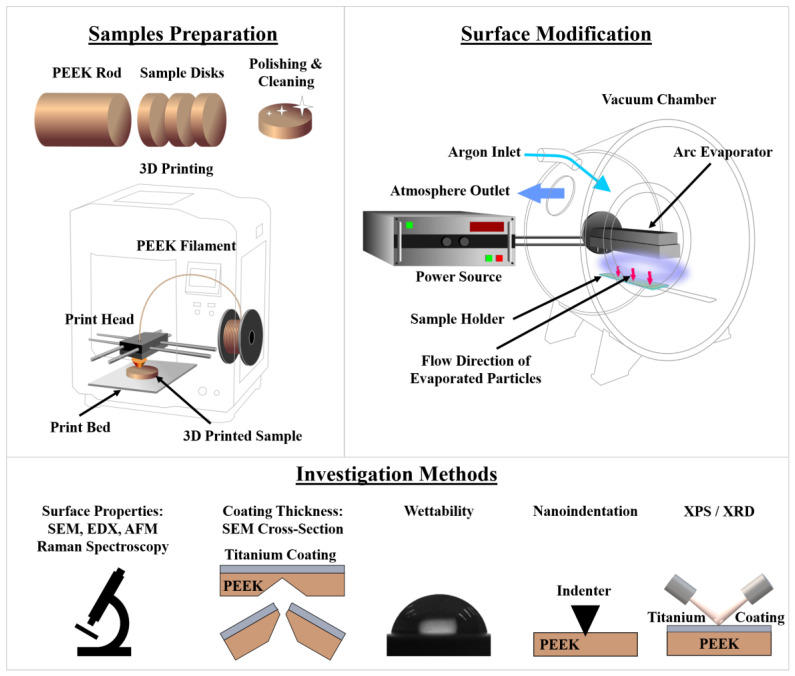
Schematic overview of the PEEK sample preparation process, the applied surface modification method, and the investigation methods used in this study.

**Figure 2 materials-16-02990-f002:**
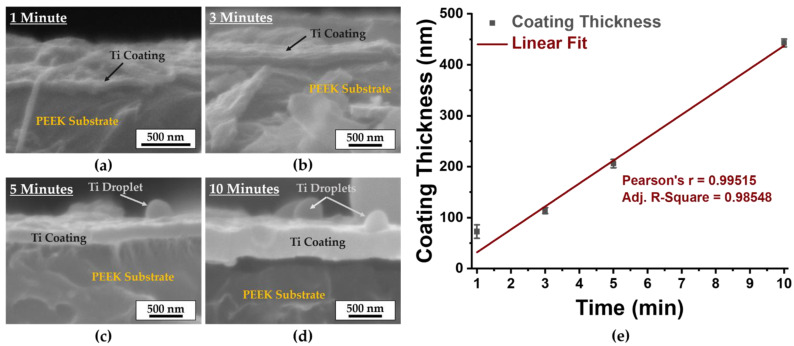
Cross-sectional SEM micrographs of surface-modified polished PEEK samples with the coating and the substrate indicated: (**a**) after 1 min, (**b**) after 3 min, (**c**) after 5 min and (**d**) after 10 min of cathodic arc evaporation. (**e**) Dependence of coatings thickness on time.

**Figure 3 materials-16-02990-f003:**
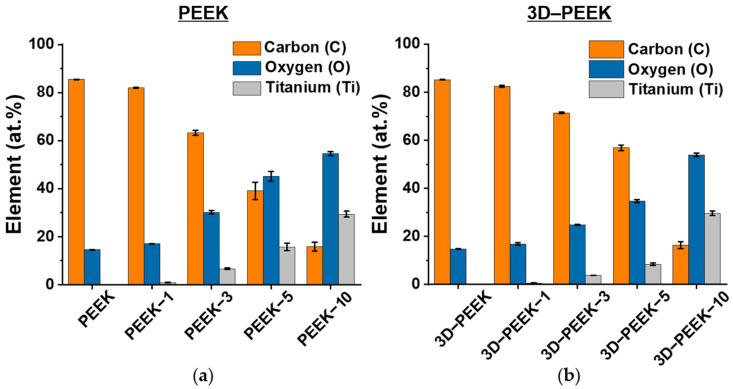
EDX analysis of the elemental composition of the examined samples that were analyzed on the content of carbon (C), oxygen (O) and titan (Ti) in atomic percentage (at.%). (**a**) Polished PEEK and (**b**) 3D–PEEK samples.

**Figure 4 materials-16-02990-f004:**
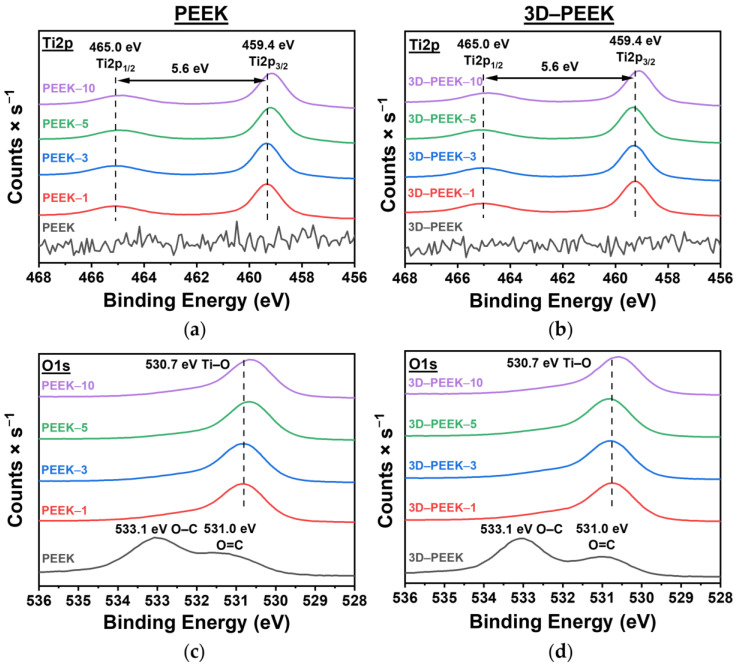
High-resolution XPS spectra for the elements titanium (Ti2p, upper row) and oxygen (O1s, lower raw) of all analyzed PEEK (first column) and 3D–PEEK samples (second column): (**a**) Ti2p spectra of the polished PEEK samples, (**b**) Ti2p spectra of the 3D–PEEK samples, (**c**) O1s spectra of the polished PEEK samples and (**d**) O1s spectra of the 3D–PEEK samples.

**Figure 5 materials-16-02990-f005:**
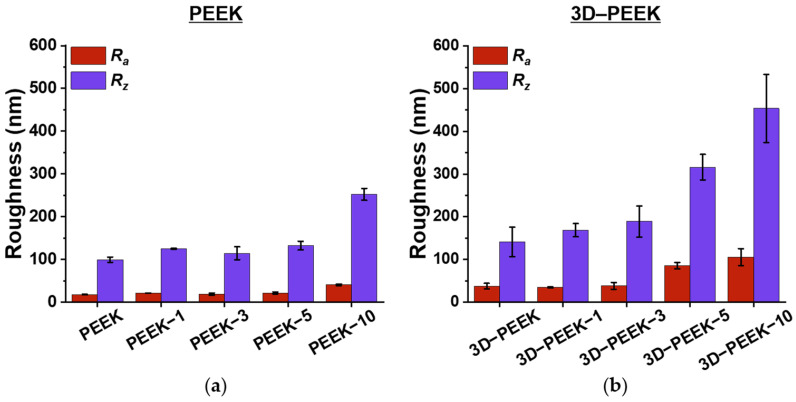
Roughness values of (**a**) polished PEEK samples and (**b**) 3D–PEEK samples are given by the arithmetic mean roughness (*R_a_*) and the average height difference (*R_z_*) measured by AFM for a sample surface area of 40 × 40 μm^2^ of each PEEK sample.

**Figure 6 materials-16-02990-f006:**
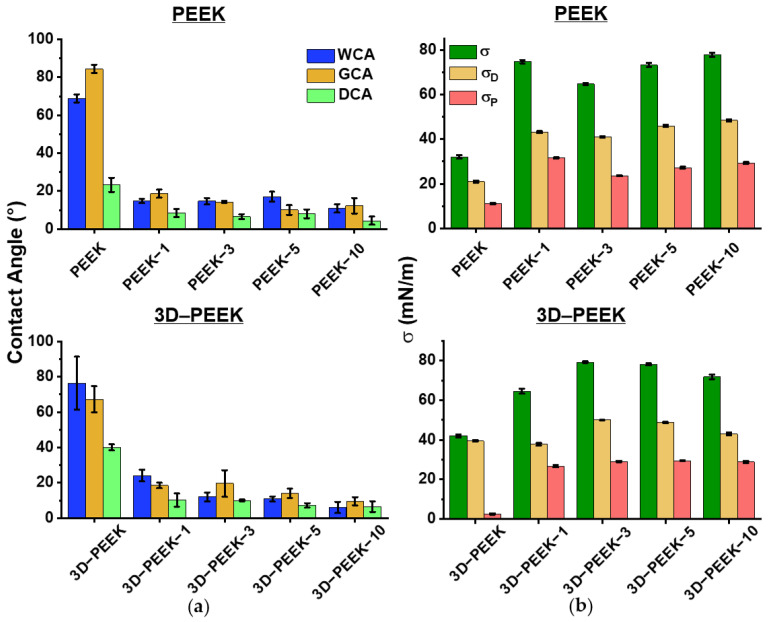
(**a**) Contact angles for all PEEK samples examined, given as water contact angles (WCA), glycerol contact angles (GCA), and diiodomethane contact angles (DCA). (**b**) The surface energy (σ) of all PEEK samples, as well as the dispersion (σ_D_) and the polar (σ_P_) components.

**Figure 7 materials-16-02990-f007:**
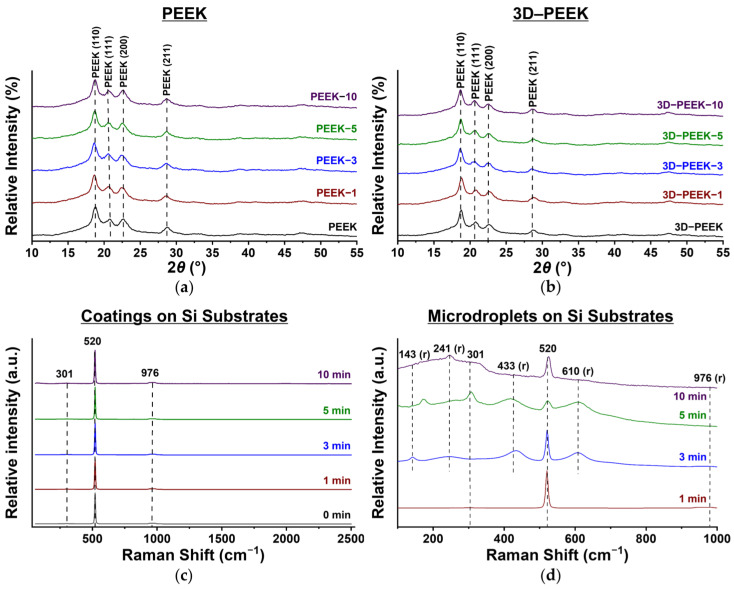
(**a**) XRD spectra of surface-modified polished PEEK and (**b**) surface-modified 3D–PEEK samples. (**c**) Raman spectra of Ti coatings formed on Si samples, and (**d**) Raman spectra of microdroplets on Si sample surfaces.

## Data Availability

Data underlying the results presented in this paper are not publicly available at this time but may be obtained from the authors upon reasonable request.

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
