# Peer review of "Improvement of the Surface Properties of Polyether Ether Ketone via Arc Evaporation for Biomedical Applications"

_materials, 2023, doi:10.3390/ma16082990_

Round 1

Reviewer 1 Report

materials-228240.

The article "Improvement of the Surface Properties of Polyether Ether Ketone via Arc Evaporation for Biomedical Applications" deals with 3D-printed and polymer extruded polyether ether ketone disc samples, which were surface-modified with titanium thin films via arc evaporation, with a significant increase in wettability. The subject is within the scope of Materials. Polyether ether ketone (PEEK) is a high-performance thermoplastic polymer that has been widely used in biomedical applications due to its excellent mechanical and chemical properties, biocompatibility, and radiolucency. However, the surface properties of PEEK, such as its wettability and adhesion, can be further improved for better performance in biomedical applications. One potential method for improving these surface properties is through arc evaporation. Arc evaporation is a physical vapor deposition technique that involves vaporizing a metal or ceramic target using an electric arc and depositing the resulting vapor onto a substrate to form a thin film. Arc evaporation is a promising technique for improving the surface properties of PEEK for biomedical applications, by deposition of thin films of a material with specific properties onto the surface of PEEK. One way is to deposit a thin film of a material with good wettability, such as titanium or titanium oxide. This can improve the surface energy of PEEK, making it more hydrophilic and thus improving its ability to bond with biological tissues or coatings. The soundness of the study is therefore correct. On the other hand, the reported increase of wettability is a priori of little interest without further studies to prove its ability to bond with biological tissues and therefore might be of less interest for readers. The abstract does not give quantitative results.

The introduction includes recent articles and briefly presents global research.

The materials and methods are properly described. The design of the study is appropriate.

The presented results have serious flaws.

The morphological study presented is not sufficient.

The surface morphology before and after coating are not presented in the SEM images. Although broken pieces of the coating can be observed, the PEEK-coating interface cannot be seen clearly, nor the boundaries of the coating, the lack of clarity is due to both low resolution and lack of contrast. In addition, the surface appears very irregular, the topographical features are much more prominent than the coating thickness.

Additional information, such as optical images of samples before and after deposition, is missing and needs to be added.

EDX spectra are not conclusive. The concomitant increase of Ti and O content seems to suggest the formation of mainly titanium oxide, but EDX does not provide information on the oxidation state of Ti atoms. The authors fail to quantify the titanium and titanium oxide content by the characterization methods used, XRD only detects PEEK and is therefore of little help, Raman cannot detect titanium metal on the other hand, so the manuscript is inconclusive about the film components.

The film properties studied are roughness and wettability, by AFM and contact angle measurements. Although quantitative values ​​of roughness are given in comparative plots, no AFM micrograph is shown (the surface morphology is therefore not shown, neither by SEM or optical images, nor by AFM images). Also, no image of contact angles is shown to illustrate changes in wetting behavior. Therefore, the study presents too many uncertainties and unresolved issues.

Furthermore, it is difficult to relate the reported results to any change in the ability of PEEK discs to bind to biological tissues after coating. The addition of such a study would be necessary.

Therefore, the conclusions are unclear. It is obvious because the authors themselves mention in the conclusions: "The obtained coatings contain titanium oxide with an amorphous structure" and then propose "the deposition of calcium phosphate on the surface of the presented titanium coatings".

Frankly, it seems like an unfinished study and needs to be improved before resubmission.

Author Response

Comment:

The article "Improvement of the Surface Properties of Polyether Ether Ketone via Arc Evaporation for Biomedical Applications" deals with 3D-printed and polymer extruded polyether ether ketone disc samples, which were surface-modified with titanium thin films via arc evaporation, with a significant increase in wettability. The subject is within the scope of Materials. Polyether ether ketone (PEEK) is a high-performance thermoplastic polymer that has been widely used in biomedical applications due to its excellent mechanical and chemical properties, biocompatibility, and radiolucency. However, the surface properties of PEEK, such as its wettability and adhesion, can be further improved for better performance in biomedical applications. One potential method for improving these surface properties is through arc evaporation. Arc evaporation is a physical vapor deposition technique that involves vaporizing a metal or ceramic target using an electric arc and depositing the resulting vapor onto a substrate to form a thin film. Arc evaporation is a promising technique for improving the surface properties of PEEK for biomedical applications, by deposition of thin films of a material with specific properties onto the surface of PEEK. One way is to deposit a thin film of a material with good wettability, such as titanium or titanium oxide. This can improve the surface energy of PEEK, making it more hydrophilic and thus improving its ability to bond with biological tissues or coatings. The soundness of the study is therefore correct. On the other hand, the reported increase of wettability is a priori of little interest without further studies to prove its ability to bond with biological tissues and therefore might be of less interest for readers. The abstract does not give quantitative results.

General answer: We thank the reviewer for his review and helpful comments. For the missing support information manuscript for the first round of review, we would like to apologize. The editor of the journal wrote to us on March 10 and informed us about this problem. Thereafter, we sent this file to the editor immediately so the editor could upload the support information manuscript to the journal system. Unfortunately, this happened after your review process for our work. We regret this circumstance and hope that all will be well with access to the support information manuscript in your second round of review. Please note that all changes in the main manuscript and the support information manuscript are marked in green.

Comment: The abstract does not give quantitative results.

Answer: We thank the reviewer for stating out this point. Quantitative results were added in the abstract as requested.

Comment: The introduction includes recent articles and briefly presents global research. The materials and methods are properly described. The design of the study is appropriate. The presented results have serious flaws.

Answer: We thank the reviewer for this judgement.

Comment: The morphological study presented is not sufficient. The surface morphology before and after coating are not presented in the SEM images. Although broken pieces of the coating can be observed, the PEEK-coating interface cannot be seen clearly, nor the boundaries of the coating, the lack of clarity is due to both low resolution and lack of contrast. In addition, the surface appears very irregular, the topographical features are much more prominent than the coating thickness.

Answer: We thank the reviewer for this observation. The SEM micrographs presented in Figure 2a – 2d show the cross-section of the samples with coatings in order to evaluate the thickness of the obtained coatings. Since the coatings are very thin, it is not possible to achieve higher sharpness and contrast at this resolution without artificial post-processing. Therefore, we only show the SEM images obtained from the microscope without post-processing. Further details on the morphology of the formed coatings can be found in the support information manuscript in figures S1 – S3.

Comment: Additional information, such as optical images of samples before and after deposition, is missing and needs to be added.

Answer: We thank the reviewer for this helpful comment. Photographs of samples before and after surface modification were added to the support information manuscript (SI Figure S2).

Comment: EDX spectra are not conclusive. The concomitant increase of Ti and O content seems to suggest the formation of mainly titanium oxide, but EDX does not provide information on the oxidation state of Ti atoms. The authors fail to quantify the titanium and titanium oxide content by the characterization methods used, XRD only detects PEEK and is therefore of little help, Raman cannot detect titanium metal on the other hand, so the manuscript is inconclusive about the film components.

Answer: We like to thank the reviewer for pointing out this issue. To address this issue, we additionally analyzed the chemical composition of the samples using XPS. The new figure 3 shows the obtained XPS spectra and a paragraph about the investigation results has been added in chapter 3 on pages 7 and 8 of the main manuscript. XPS survey spectra has been added to the support information manuscript in SI Figure S5.

Comment: The film properties studied are roughness and wettability, by AFM and contact angle measurements. Although quantitative values of roughness are given in comparative plots, no AFM micrograph is shown (the surface morphology is therefore not shown, neither by SEM or optical images, nor by AFM images). Also, no image of contact angles is shown to illustrate changes in wetting behavior. Therefore, the study presents too many uncertainties and unresolved issues.

Answer: We thank the reviewer for this helpful comment. SEM and AFM-images of the investigated samples are shown in support information manuscript (SI Figure S1, S2 and S3). The requested images of the measured contact angles were added in the support information manuscript in SI Figure S7. It should be noted that the changes in contact angles correspond well with the changes in roughness within the Wenzel-Deryagin model (Figure 4 and Figure 5).

Comment: Furthermore, it is difficult to relate the reported results to any change in the ability of PEEK discs to bind to biological tissues after coating. The addition of such a study would be necessary.

Answer: We would like to thank the reviewer for this comment. For this article, we have focused on the feasibility of fabricating titanium coatings on two different PEEK substrates. Our aim was to study possible differences and the physico-chemical properties of these coatings on PEEK substrates fabricated by cathodic arc evaporation. Additional biological studies would take the manuscript out of focus and therefore were not planned. In addition, when planning the study, we had to keep in mind that we had only a short time to conduct the study, as we were invited by the journal to contribute to a special issue. To continue research in this area, we are already planning to conduct a larger in vitro comparison study with different surface modified PEEK samples and different coating processes. This will be the subject of a separate article for the reasons mentioned. In more detail, this planned article will be a comparative in vitro study of PEEK samples surface-modified with ultra-thin films of titanium oxide and zirconium oxide fabricated by DC magnetron sputtering (our previous article [1]) and PEEK samples surface-modified with titanium thin films fabricated via arc evaporation (current article). However, as indirect evidence of the change in the ability of PEEK to bind to biological tissues after the application coatings, there are research results from other authors, which we had already indicated in the introduction (references no. 19, 22–24).

Comment: Therefore, the conclusions are unclear. It is obvious because the authors themselves mention in the conclusions: "The obtained coatings contain titanium oxide with an amorphous structure" and then propose "the deposition of calcium phosphate on the surface of the presented titanium coatings".

Answer: We thank the reviewer for stating out this point. In the conclusion, the phrase of "The obtained coatings contain titanium oxide with an amorphous structure" refers to the description of the coatings formed. Likewise, the phrase "the deposition of calcium phosphate on the surface of the presented titanium coatings" was part of a sentence that stated that the coatings formed could be used as a sublayer for subsequent deposition of calcium phosphate coatings. In accordance with the this comment, we have reformulated these phrases to make them more understandable.

Comment: Frankly, it seems like an unfinished study and needs to be improved before resubmission.

Answer: We thank the reviewer for this judgement. We regret that the reviewer has such an impression of our article. We believe that this impression has not been fully caused by the main manuscript itself, but in the lack of the support information manuscript, which unfortunately was not available during the review process of the reviewer. The article has been improved in accordance with the comments of all reviewers and we will try to ensure that the newest version of the support information manuscript is also available for the second revision round.

Reference:

  1. Akimchenko, I.O.; Rutkowski, S.; Tran, T.H.; Dubinenko, G.E.; Petrov, V.I.; Kozelskaya, A.I.; Tverdokhlebov, S.I. Polyether Ether Ketone Coated with Ultra-Thin Films of Titanium Oxide and Zirconium Oxide Fabricated by DC Magnetron Sputtering for Biomedical Application. Materials 2022, 15, 8029, doi:10.3390/MA15228029/S1.

Reviewer 2 Report

The work of Fedotkin et al. reports on the changes and improvement in the surface properties of polyether ether ketone with a titanium coating deposited by arc evaporation. The manuscript is well structured and the data supports the conclusion. The topic of the manuscript is a good fit for the special issue it is submit to ("Nanostructured Materials for Biomedical Applications"), nevertheless to note that the layers are only nanostructured as a result of the titanium layer deposition which presents a "microdroplet" morphology. The following modifications can be included into the revision, as discussed below. I encourage the authors to improve their work and resubmit (please also check that you upload the supporting information).

1. The supporting information file is missing (only the cover letter to the editor is attached).

2. Therefore, it is impossible to advise with respect to the EDX data of the Ti layer with increasing thickness. However, with respect to the presence of titanium dioxide and of C, author can include an XPS measurement of a Ti coated PEEK. As XPS is surface sensitive techniques and measures up to 10nm, authors can that C is only from the PEEK (in this conditions only adventitious C should be present), and they will be able to see the chemical state of Ti. To keep in mind that typically Ti is easily oxidized in air, and for Ti metal (even polished and measured immediatly after), there is still native oxide formed and confirmed by XPS.

3. Figure 3,4. Authors can include some legend/note on the figure directly to emphasize which is for PEEK and which for 3D-PEEK. Is there a difference in the thickness of the layers when deposited on the two different PEEK substrates? For example, for the at.% of the 5min Ti deposition there is a significant difference in the composition observed from EDX (PEEK - Ti at around 15at%, 3D-PEEK at 7-8at%; values are read from the plot in Figure 3 as SI were not available). The PEEK and 3D-PEEK have similar at.%, and also part of the O is from this substrate.

4. For the roughness data, the 3D-PEEK substrate is sligthly rougher than the PEEK one. Why is the roughness and the average height difference increasing so much only for the 3D-PEEK samples after the Ti coating was deposited for 5min and 10min?

5. ln 310 with respect to the XRD data "Therefore, polished sample surfaces can be used as a model for studying the properties of 3D-printed scaffolds surface-modified by cathodic arc evaporation". Completely true, from the point of view of the XRD data. However, from the data shown into the manuscript (without SI, as it is not available to the reviewer at the current time), from the roughness data there is a significant difference in the roughness and the average heigh difference in between the PEEK and 3D-PEEK with 5 or 10min deposition. Which means that there is a different "micro- and nanoscale relief" on the samples with it being more pronounced on the 3D-PEEK. Authors should discuss also these differences and how this affects using the PEEK samples as a model.

6. For the XRD - completely agree with the authors, that no Ti peak could be detected in this conditions. Authors can modify the XRD measurements settings for a higher aquisition time and specifically for measuring thin films.

7. Minor things: a) please include the a),b) legends near the top part of the figures; b) minor English and typo corrections are needed.

Author Response

Comment: The work of Fedotkin et al. reports on the changes and improvement in the surface properties of polyether ether ketone with a titanium coating deposited by arc evaporation. The manuscript is well structured and the data supports the conclusion. The topic of the manuscript is a good fit for the special issue it is submit to ("Nanostructured Materials for Biomedical Applications"), nevertheless to note that the layers are only nanostructured as a result of the titanium layer deposition which presents a "microdroplet" morphology. The following modifications can be included into the revision, as discussed below. I encourage the authors to improve their work and resubmit (please also check that you upload the supporting information).

General answer: We thank the reviewer for his review and helpful comments. Please note that all changes in the main manuscript and the support information manuscript are marked in green.

Comment 1: The supporting information file is missing (only the cover letter to the editor is attached).

Answer: We thank the reviewer for pointing out this issue. For the missing support information manuscript for the first round of review, we would like to apologize. The editor of the journal wrote to us on March 10 and informed us about this problem. Thereafter, we sent this file to the editor immediately so the editor could upload the support information manuscript to the journal system. Unfortunately, this happened after your review process for our work. We regret this circumstance and hope that all will be well with access to the support information manuscript in your second round of review.

Comment 2: Therefore, it is impossible to advise with respect to the EDX data of the Ti layer with increasing thickness. However, with respect to the presence of titanium dioxide and of C, author can include an XPS measurement of a Ti coated PEEK. As XPS is surface sensitive techniques and measures up to 10nm, authors can that C is only from the PEEK (in this conditions only adventitious C should be present), and they will be able to see the chemical state of Ti. To keep in mind that typically Ti is easily oxidized in air, and for Ti metal (even polished and measured immediatly after), there is still native oxide formed and confirmed by XPS.

Answer: We thank the reviewer for this very helpful comment. XPS study of all samples has been conducted. The high-resolution XPS spectra were added in the main manuscript in the new Figure 4. In addition, the summary spectra and a table with the values of the content of the detected elements were included in the information manuscript in SI Figure S5 and SI Table S2. Analysis of the high-resolution XPS Ti2p spectra shown in Figure 4 revealed that the distance of about 5.6 eV between the peaks indicates the oxidation state of Ti4+ [1], which is consistent with that in TiO2. It should be noted that no TiO2 phase was found in the XRD studies, indicating an amorphous structure.

Comment 3: Figure 3,4. Authors can include some legend/note on the figure directly to emphasize which is for PEEK and which for 3D-PEEK. Is there a difference in the thickness of the layers when deposited on the two different PEEK substrates? For example, for the at.% of the 5min Ti deposition there is a significant difference in the composition observed from EDX (PEEK - Ti at around 15at%, 3D-PEEK at 7-8 at%; values are read from the plot in Figure 3 as SI were not available).

Answer: We thank the reviewer for this helpful suggestion. The sample type is now indicated in a better was in all relevant figures. The analysis of the SEM micrographs of the coating cross sections of the two different PEEK samples (PEEK and 3D-PEEK, see Figure 2 and SI Figure S1), whose coatings were prepared in the same mode of cathodic arc evaporation, shows an identical thickness of the coatings.

The differences in composition (PEEK-5 and 3D-PEEK-5) detected by EDX may be due to several factors:

1) Sorbed carbon from the sample environment may promote the increase of carbon content;

2) The sorbed water molecules from the environment can also influence the increase of oxygen content;

3) The angle at which the electron beam impinges on the surface of the samples under investigation during SEM-EDX examination. The closer the angle is to 90°, the lower the reduced film thickness;

4) The number of microdroplets formed during the formation of the coatings. EDX mapping showed a high Ti content in the composition of the microdroplets, while the oxygen content remained the same.

We have added three columns to SI Table S1: 1) the amount of oxygen bound to carbon at a ratio of 0.16/1 as for PEEK [2]; 2) the amount of oxygen bound to Ti ([O bonded to C] - C (second column)); 3) the ratio Ti /O. Moreover, a paragraph about it has been added to main manuscript on page 8.

Based on the values obtained, the Ti/O ratio for 3D PEEK samples surface-modified for 3 and 5 min were 0.31 and 0.34, respectively. For polished PEEK samples surface-modified for 3 and 5 min, the Ti/O ratio was 0.33 and 0.40, respectively. This shows that the results are quite close, taking into account the above mentioned Ti/O ratios.

Comment 4: For the roughness data, the 3D-PEEK substrate is sligthly rougher than the PEEK one. Why is the roughness and the average height difference increasing so much only for the 3D-PEEK samples after the Ti coating was deposited for 5min and 10min?

Answer: We thank the reviewer for addressing these interesting questions. It is well known that the roughness of the substrate affects the development of surface roughness during the deposition of thin films [3]. The initial surface roughness of 3D PEEK samples results in the formation of a rapidly profilerating roughness profile due to the effect of geometric shadowing [4–6]. In particular, the irregularities on 3D-PEEK samples mask other parts of the growing surface from the emerging particle stream during the arc evaporation process. In turn, the peaks of the surface irregularities are exposed to impinging particles from all directions and thus increase in size. This effect is most pronounced when the coating is sufficiently thick, i.e. after a treatment time of 5 and 10 minutes. The droplet fraction of the evaporated target also contributes additionally to the roughness. Increasing the deposition time leads to an increase in the number of microdroplets deposited on the sample surfaces, as can be seen in SI Figures S1 – S3. This explanation has been added to the main manuscript in a paragraph on pages 8 and 9.

 Comment 5: ln 310 with respect to the XRD data "Therefore, polished sample surfaces can be used as a model for studying the properties of 3D-printed scaffolds surface-modified by cathodic arc evaporation". Completely true, from the point of view of the XRD data. However, from the data shown into the manuscript (without SI, as it is not available to the reviewer at the current time), from the roughness data there is a significant difference in the roughness and the average heigh difference in between the PEEK and 3D-PEEK with 5 or 10min deposition. Which means that there is a different "micro- and nanoscale relief" on the samples with it being more pronounced on the 3D-PEEK. Authors should discuss also these differences and how this affects using the PEEK samples as a model.

Answer: We thank the reviewer for this comment. According to the SEM, XPS, XRD and Raman spectroscopy data, the coatings on the surface of PEEK and 3D PEEK substrates have the same characteristics. For surface roughness and wettability, the numerical values are slightly different. At the same time, as the deposition time of the cathodic arc evaporation process increases, the roughness and wettability of the coatings also increase. From all this, it can be concluded that polished sample surfaces can generally be used as a model for studying the properties of 3D-printed samples whose surfaces have been modified by cathodic arc evaporation. This explanation has been added to the conclusion.

Comment 6: For the XRD - completely agree with the authors, that no Ti peak could be detected in this conditions. Authors can modify the XRD measurements settings for a higher aquisition time and specifically for measuring thin films.

Answer: We thank the reviewer for this suggestion. The samples PEEK-10 and 3D-PEEK-10 were measured again. The acquisition time (preset time) of the re-measurement has been increased four times to 4 seconds. To shorten the total time for measuring a sample, we selected a scan range of 10°-43° (the angular range where the main peaks of titanium and titanium oxide are located), and the scan distance was increased to 0.03. Measurements were made with the grazing incidence of the beam at 3°. Phases of titanium and titanium oxide were not detected. In the XRD diffractograms, only the peaks of the PEEK substrate are present. Thus, increasing the acquisition time had no effect on the diffractograms. The results are shown below (please see attached answer file).

Figure 1. XRD diffractogram of the PEEK-10 sample obtained at an aquisition time of 4s.

Figure 2. XRD diffractogram of the 3D-PEEK-10 sample obtained at an aquisition time of 4s.

Comment 7: Minor things: a) please include the a),b) legends near the top part of the figures; b) minor English and typo corrections are needed.

Answer: We thank the reviewer for these suggestions. We have not changed the position of a), b), ... have not been changed, because we have followed the MDPI Material template for the figure details, as it can be seen in the actual template on page 3: https://www.mdpi.com/files/word-templates/materials-template.dot We rechecked the manuscript for English and typos and corrected the ones we found.

References:

  1. Zhang, Y.; Chen, K.; Zhang, J.; Huang, K.; Liang, Y.; Hu, H.; Xu, X.; Chen, D.; Chang, M.; Wang, Y. Dense and Uniform Growth of TiO2 Nanoparticles on the Pomelo-Peel-Derived Biochar Surface for Efficient Photocatalytic Antibiotic Degradation. J Environ Chem Eng 2023, 11, 109358, doi:10.1016/J.JECE.2023.109358.
  2. Pawson, D.J.; Ameen, A.P.; Short, R.D.; Denison, P.; Jones, F.R. An Investigation of the Surface Chemistry of Poly(Ether Etherketone). I. The Effect of Oxygen Plasma Treatment on Surface Structure. Surface and Interface Analysis 1992, 18, 13–22, doi:10.1002/SIA.740180104.
  3. Pelliccione, M.; Lu, T.-M. Evolution of Thin Film Morphology. Springer: New York, NY, USA 2008.
  4. Yao, J.H.; Guo, H. Shadowing Instability in Three Dimensions. Phys Rev E 1993, 47, 1007, doi:10.1103/PhysRevE.47.1007.
  5. Drotar, J.T.; Zhao, Y.; Lu, T.; Wang, G. Surface Roughening in Shadowing Growth and Etching in 2 + 1 Dimensions. Phys Rev B 2000, 62, 2118, doi:10.1103/PhysRevB.62.2118.
  6. Turkin, A.A.; Pei, Y.T.; Shaha, K.P.; Chen, C.Q.; Vainshtein, D.I.; De Hosson, J.T.M. On the Evolution of Film Roughness during Magnetron Sputtering Deposition. J Appl Phys 2010, 108, 094330, doi:10.1063/1.3506681.

Reviewer 3 Report

In this manuscript, the authors compared the surface properties of titanium-coated PEEK and 3D-printed PEEK samples by cathodic arc evaporation in terms of surface roughness, surface morphology, wettability, composition, etc. The results were well presented and discussed. Therefore, this paper can be accepted in the current form.

Author Response

In this manuscript, the authors compared the surface properties of titanium-coated PEEK and 3D-printed PEEK samples by cathodic arc evaporation in terms of surface roughness, surface morphology, wettability, composition, etc. The results were well presented and discussed. Therefore, this paper can be accepted in the current form.

Answer: We thank the reviewer for the friendly judgment about our work.

Round 2

Reviewer 1 Report

The support information missing for the first round of review was indeed a problem. Most of the missing information can be found in the support document.
I cannot see the advantages of presenting the results of the article mainly in the supporting info manuscript. Said that, once taken into account the supplementary information, the work appears much more a complete study. The morphological study presented is much improved. Photographs of samples before and after surface modification were added to the support information manuscript (the scale bars in Figure S2 are wrong, please correct them).
The authors have quantified the titanium and titanium oxide content by analyzing the chemical composition of the samples using XPS, the obtained XPS spectra and a paragraph about the investigation results has been added in the main manuscript. XPS survey spectra and EDX mapping images can be found in the support information.
SEM and AFM images of the investigated samples are shown in the SI.

The requested images of the measured contact angles were added in the SI.
In the conclusion, the phrase "the deposition of calcium phosphate on the surface of the presented titanium coatings" was reformulated.

The article has been improved in accordance with the comments and only minor revisions are now necessary.

Author Response

The support information missing for the first round of review was indeed a problem. Most of the missing information can be found in the support document.
I cannot see the advantages of presenting the results of the article mainly in the supporting info manuscript. Said that, once taken into account the supplementary information, the work appears much more a complete study. The morphological study presented is much improved. Photographs of samples before and after surface modification were added to the support information manuscript (the scale bars in Figure S2 are wrong, please correct them).
The authors have quantified the titanium and titanium oxide content by analyzing the chemical composition of the samples using XPS, the obtained XPS spectra and a paragraph about the investigation results has been added in the main manuscript. XPS survey spectra and EDX mapping images can be found in the support information.
SEM and AFM images of the investigated samples are shown in the SI.

The requested images of the measured contact angles were added in the SI.
In the conclusion, the phrase "the deposition of calcium phosphate on the surface of the presented titanium coatings" was reformulated.

The article has been improved in accordance with the comments and only minor revisions are now necessary.

General answer: We like to thank the reviewer for reviewing our work.

Comment: I cannot see the advantages of presenting the results of the article mainly in the supporting info manuscript.

Answer: We thank the reviewer for sharing his opinion. Please let us state our opinion. To divide a research paper into a main manuscript and a support information manuscript has some benefits. First, overloading a paper can be avoided, as it is a reason for rejecting a paper. This point is especially important for publications in high impact factor journals. For example, a paper with 15 figures and 8 tables is mostly overloaded and leads to some effects known in perception psychology: - the reader can much more easily lose the important "red thread" within a manuscript, which leads to frustration, - it promotes the confusion of a manuscript, which leads to the fact that already in the middle of the manuscript you sometimes do not know what was shown in the first figures. Second, the psychological points should be taken into account in a good written paper. The support information manuscript is therefore a very helpful tool given by the most of the journals. Therefore, it is a trend since some years to design the main manuscript only with the most important figures and to put the rest in the support information manuscript, which can be observed especially in high impact factor journals. Our research group is therefore open about newer trends in paper writing.

Comment: Said that, once taken into account the supplementary information, the work appears much more a complete study.

Answer: We thank the reviewer for this statement. We can only apologize for this circumstance that the support information manuscript was not available for the first round of your revision. The uploading author will try to avoid this error in the future. Additionally, we would like to thank the journal managing editor who noticed our mistake, wrote us a separate email and uploaded the missing file.

Comment: The scale bars in Figure S2 are wrong, please correct them.

Answer: We thank the reviewer for pointing out this issue. Indeed, we got confused with the correct value for the scale bars. This error has been corrected and SI Figure S2 has been replaced with the corrected figure.

Please see attached file “materials-2282407_R2_support_information_unmarked”.
